# Prevalence of Human Papillomavirus Types 16/18 and Effect of Vaccination among Japanese Female General Citizens in the Vaccine Crisis Era

**DOI:** 10.3390/v15010159

**Published:** 2023-01-04

**Authors:** Tadaichi Kitamura, Motofumi Suzuki, Kazuyoshi Shigehara, Kazuko Fukuda, Taeko Matsuyama, Haruki Kume

**Affiliations:** 1Japanese Foundation of Sexual Health Medicine, Tokyo 113-0034, Japan; 2Department of Urology, Tokyo Metropolitan Bokutoh Hospital, Tokyo 130-8575, Japan; 3Department of Urology, Faculty of Medicine, Kanazawa University, Kanazawa 920-8641, Japan; 4Department of Nursing, Tachikawa Faculty of Nursing, Tokyo Healthcare University, Tachikawa 190-8590, Japan; 5Department of Urology, Graduate School of Medicine, The University of Tokyo, Tokyo 113-8655, Japan

**Keywords:** human papillomavirus, epidemiology, Japanese female general citizens, cross-sectional study, self-sampling

## Abstract

The Japanese government withdrew its recommendation for human papillomavirus (HPV) vaccination in June 2013 and resumed it in April 2022. This period is known as the vaccine crisis in Japan. This study aimed to elucidate the prevalence and genotype distribution of HPV among Japanese female citizens, and the effect of vaccination against HPV-16/18 in the era of the vaccine crisis. We recruited Japanese female citizens and asked them to provide self-collected samples from the vaginal wall using cotton swabs for HPV genotyping. Furthermore, we collected the participants’ characteristics, including lifestyle and experience of vaccination against HPV, to determine the significant association with HPV infection. HPV-16/18 positivity was found in 5.6% (115/2044) of participants. The highest vaccination rate was observed in the age group of 20–24 years (60.6%), whereas the lowest HPV-16/18 positivity was observed in the age group of 45–49 years (2.8%), followed by the age group of 20–24 years (4.0%). Experience with HPV vaccination significantly reduced the risk of HPV-16/18 infection (adjusted odds ratio, 0.047; 95% confidence interval, 0.011–0.196). Vaccinated women were much less likely to be infected by HPV-16/18, regardless of the HPV vaccine type or the vaccination dose.

## 1. Introduction

According to a report by GLOBOCAN [1], 604,127 women were diagnosed with cervical cancer worldwide, and 341,831 women died of this disease. Cervical cancer is the fourth most common cancer in women, ranking after breast, colorectal, and lung cancers [1]. According to recent statistics in Japan [2], 10,879 women were diagnosed with cervical cancer in 2019, and 2887 women died of this disease in 2020. Cervical cancer is the 12th most common cancer in Japanese women, ranking after breast, colorectal, lung, stomach, pancreas, endometrial, malignant lymphoma, thyroid, ovarian, skin, and liver cancers. Notably, in the age group of 30–39 years, cervical cancer is the second most common cancer in young Japanese women, ranking after breast cancer [3]. A majority of cervical cancer is caused by the human papilloma virus (HPV) infection. There are six licensed vaccines (three bivalent, two quadrivalent, and one nonavalent vaccine), and all of them are highly efficacious in preventing infection with HPV-16/18, which are responsible for approximately 70% of cervical cancer cases globally [4]. The nonavalent vaccine protects against five additional high-risk genotypes of HPV (HPV-31/33/45/52/58), which cause a further 20% of cervical cancers [5].

In Japan, the bivalent HPV vaccine (target genotypes: HPV-16/18), quadrivalent HPV vaccine (target genotypes: HPV-6/11/16/18), and nonavalent HPV vaccine (target genotypes: HPV-6/11/16/18/31/33/45/52/58) were available with the approval of the Ministry of Health, Labour and Welfare of Japan in December 2009, August 2011, and February 2021, respectively. In Japan, public subsidies for HPV vaccination started in 2010, and HPV vaccination was a routine vaccination based on the Immunization Law in April 2013; however, the active recommendation of HPV vaccination has been suspended since June 2013 due to various symptoms, including chronic pain and motor impairment reported after the vaccination. Additionally, the HPV vaccination rate among girls born between 1994 and 1999 who were eligible for vaccination during the public subsidy was approximately 70%. In contrast, the rate among girls born in 2000 and later decreased drastically, and the rate among girls born in 2002 and later was <1% [6]. The suspension of HPV vaccination in Japan is called a vaccine crisis [7]. The vaccine crisis from 2013 to 2019 is predicted to result in an additional 24,600–27,300 new cases and 5000–5700 deaths from cervical cancer over the lifetime of cohorts born between 1994 and 2007, compared with the number of new cases and deaths that would have occurred if coverage had remained at approximately 70% since 2013. However, coverage restoration in 2020, including catch-up vaccination for missed cohorts, could prevent 14,800–16,200 of these cases and 3000–3400 deaths [8].

The Japanese government resumed a recommendation for HPV vaccination in April 2022. The Ministry of Health, Labour and Welfare in Japan also started a catch-up vaccination for women born from April 1997 to April 2006 who lost an opportunity for HPV vaccination in the era of the vaccine crisis. The bivalent, quadrivalent, and nonavalent HPV vaccines were available in Japan, and HPV-16/18 were common genotypes covered by these vaccines; however, studies concerning the prophylactic effect of these vaccines against HPV-16/18 infection have been very scarce in Japan.

Several epidemiological studies have been conducted on Japanese female citizens who have undergone cervical cancer screening to assess the prevalence of HPV-16/18 positivity. In contrast, studies on female Japanese general citizens are limited [9,10,11]. Therefore, this study aimed to elucidate the prevalence and genotype distribution of HPV among Japanese female citizens and the effect of vaccination against HPV-16/18 in the era of the vaccine crisis.

## 2. Materials and Methods

### 2.1. Material Collection

We previously reported the prevalence and risk factors of HPV among 1003 Japanese female general citizens recruited from April 2017 to March 2020 [12]. However, in the present study, we focused on the prevalence of HPV-16/18 and the prophylactic effect of vaccination against HPV-16/18 in a larger number of Japanese female general citizens. We continued nationwide recruitment until March 2022 through the Japanese Foundation for Sexual Health Medicine (https://www.jfshm.org/, accessed on 23 October 2022). When a woman agreed to participate in the survey at our homepage, we sent her a cotton swab with a plastic tube container and a printed survey form to collect information on age at study entry, educational attainment, smoking status, number of lifetime sex partners, age of coitarche, marital status, divorce experience, number of children, commercial sex work experience, present history of sexually transmitted infection (STI), past history of STI, and HPV vaccination status. Except during the menstrual period, participants were asked to self-collect a vaginal sample by rubbing a cotton swab on the vaginal wall several times and placing the sample in a plastic tube container.

### 2.2. HPV Detection and Genotyping Procedures

Self-collected vaginal samples were transported to LSI Medience Corporation (Tokyo, Japan), and HPV DNA and β-globin were detected using the GENOSEARCH HPV31 kit (Medical and Biological Laboratory, Nagoya, Japan) according to the manufacturer’s instructions [13]. Briefly, DNA was extracted from self-collected vaginal samples using the Smitest EX&D (Medical and Biological Laboratory, Nagoya, Japan). The GENOSEARCH HPV31 kit employs a polymerase chain reaction with the reverse sequence-specific oligonucleotide (PCR-rSSO) probe targeting the L1 gene of HPV DNA. The PCR-rSSO method combines genotype-specific multiplex PCR and Luminex^®^ technology that enables simultaneous measurement of co-infections of HPV using fluorescent beads. The PCR products with fluorescent beads were read by the Luminex^®^ 100xMAP flow cytometry dual-laser system (Luminex, Austin, TX, USA). The test kit can detect the following 31 genotypes of HPV DNA: high-risk (types 16, 18, 26, 31, 33, 35, 39, 45, 51, 52, 53, 56, 58, 59, 66, 68, 70, 73, and 82) and low-risk (types 6b, 11, 42, 44, 54, 55, 61, 62, 71, 84, 90, and CP6108) [14]. Additionally, it contains a probe for detecting human β-globin to ensure that the self-collected vaginal samples contain sufficient cellular components. The Ethics Committees of the Japanese Foundation for Sexual Health Medicine approved all protocols and assessments used in this study (JFSHM No. 1 and JFSHM No. 5). Written informed consent was obtained from all participants.

### 2.3. Statistical Methods

Statistical analyses were performed using the JMP Pro version 16.1.0 (SAS, Cary, NC, USA). The positivity of HPV-16/18 and HPV vaccination by the age group were compared using the chi-square test. The chi-square test was also used to compare the status of HPV vaccination and participant characteristics, except for age at study entry. The ages of the unvaccinated and vaccinated participants were compared using the Wilcoxon rank-sum test. The association between participant characteristics and HPV infection was analyzed using logistic regression analysis. In the multivariate analysis, all covariates were included in the risk calculation. In the post hoc analysis, we compared the characteristics of participants who initiated sex debut at the age of 20 or older and those who initiated sex debut younger than 20 years. We used the Wilcoxon rank-sum test to compare ages. We also used the chi-square test to compare the remaining characteristics. Statistical significance was set at *p* < 0.05.

## 3. Results

### 3.1. Study Participants

The study population comprised 2095 female Japanese citizens aged 16–75 years. After excluding 51 participants (21 virgin participants, 19 participants with duplications of data, 7 participants of unknown age, and 4 participants who did not undergo β-globin detection), 2044 participants were included in the analysis. Table 1 shows the characteristics of the participants. The vaccination status of 11 participants was unknown. Of the remaining 2033 participants, the types of HPV vaccine used were bivalent, quadrivalent, nonavalent, and unknown in 161, 74, 14, and 205 participants, respectively. Furthermore, the doses of vaccination were once in 18, twice in 62, three times in 115, and unknown in 259 participants. Therefore, we employed vaccination experience in the logistic regression analysis, regardless of the HPV vaccine type or vaccination dose.

### 3.2. Status of HPV Vaccination and Participant Characteristics

We compared the HPV vaccination status with participant characteristics (Table 2). Except for the number of lifetime sex partners, the experience of commercial sex work, and the past history of STI, parameters were significantly different between unvaccinated and vaccinated participants.

### 3.3. Distribution of HPV Genotypes and Coverage of HPV Vaccination by Age Groups

Of the 2044 participants, 897 (43.9%), 674 (33.0%), 594 (29.1%), and 371 (18.2%) had at least one, high-risk, low-risk, and both HPV genotypes, respectively. The most frequently detected high-risk HPV genotype was HPV-52 (9.1%, 187/2044), followed by HPV-53 (5.6%, 114/2044), HPV-56 (5.4%, 110/2044), HPV-58 (3.9%, 80/2044), HPV-16 (3.9%, 79/2044), HPV-51 (3.8%, 77/2044), HPV-66 (3.8%, 77/2044), HPV-39 (3.7%, 75/2044), HPV-68 (3.6%, 74/2044), HPV-82 (3.1%, 63/2044), HPV-59 (2.7%, 56/2044), and HPV-18 (2.0%, 40 /2044), among others (Figure 1). Of 897 participants, 386 (43.0%) participants were infected by a single genotype of HPV, whereas the remaining 511 (57.0%) participants were infected by multiple genotypes of HPV (Figure 2).

Overall, the positivity of HPV-16/18 was 5.6% (115/2044): 9.1% (3/33), 4.0% (17/423), 7.3% (39/531), 6.3% (25/398), 4.8% (13/270), 5.7% (11/193), 2.8% (3/109), and 4.6% (4/87) in the age groups of 16–19, 20–24, 25–29, 30–34, 35–39, 40–44, 45–49, and 50 years and older, respectively (*p* = 0.288). Although the types of HPV vaccine (bivalent, quadrivalent, or nonavalent), doses of vaccination, and age of vaccination were not fully recorded, 22.3% (454/2033; missing, n = 11) of participants were vaccinated with HPV; 16.1% (5/31), 60.6% (254/419), 23.3% (123/528), 9.6% (38/397), 8.2% (22/270), 4.2% (8/192), 3.7% (4/109), and 0% (0/87) in the age groups of 16–19, 20–24, 25–29, 30–34, 35–39, 40–44, 45–49, and 50 years or older, respectively (*p* < 0.001). The highest vaccination rate was observed in the age group of 20–24 years (60.6%), whereas the lowest HPV-16/18 positivity was observed in the age group of 45–49 years (2.8%), followed by the age group of 20–24 years (4.0%).

### 3.4. Association between Characteristics of the Participants and Risk of HPV Infection

In multivariate analyses, the number of lifetime sex partners (≥6 persons), age of sex initiation (≥20 years of age), unmarried status, and current and past history of STI were significantly associated with high-risk HPV infection (Table 3).

Meanwhile, the number of lifetime sex partners (≥6 persons), current and past history of STI, and HPV vaccination status were significantly associated with HPV-16/18 infection. Additionally, the experience of vaccination against HPV, regardless of HPV vaccine type or vaccination dose, significantly reduced the risk of HPV-16/18 infection (odds ratio, 0.047; 95% confidence interval, 0.011–0.196) (Table 4).

We focused on the changes in the odds ratios regarding the association between the age of coitarche and high-risk HPV or HPV-16/18 infections. The odds ratios of the age at coitarche in the univariate analysis were <1. Therefore, this implies that women who initiated sex debut at the age of 20 or older were less likely to be infected with high-risk HPV or HPV-16/18. However, these odds ratios were above 1 in the multivariate analyses. This implies that women who initiated sex debut at the age of 20 or older were more likely to be infected with high-risk HPV or HPV-16/18. Furthermore, the participants who initiated sex debut at the age of 20 or older were older than those who initiated sex debut younger than 20 years (median (interquartile range); 30 (26–37) years old vs. 29 (24–36) years old, *p* < 0.001). The former had a higher proportion of educational period >12 years (91.3%; 81.9%, *p* < 0.001), a higher proportion of never-smokers (84.1%; 69.5%, *p* < 0.001), a higher proportion of lifetime sex partners with 1–5 persons (72.5%; 34.4%, *p* < 0.001), a lower experience of divorce (5.6%; 9.5%, *p* < 0.001), a lower rate of commercial sex work experience (4.5%; 11.6%, *p* < 0.001), and lower rates of a past history of STI (23.3%; 36.8%, *p* < 0.001) than the latter. However, the HPV vaccination rate of the former group was significantly lower than that of the latter (23.3% vs. 36.8%; *p* < 0.001).

## 4. Discussion

According to epidemiological data from Japan, 40%–50% of cervical cancers are caused by HPV-16 and 20%–30% by HPV-18, followed by HPV-52, HPV-58, HPV-31, and HPV-33, which are detected in cervical cancer. Specifically, 60%–70% of cervical cancers are caused by HPV-16 and HPV-18 [15,16]. All HPV vaccines (bivalent, quadrivalent, and nonavalent) available in Japan cover both HPV-16 and HPV-18. According to our data, vaccinated women had the following characteristics: younger, a longer educational status, a higher proportion of never smokers, earlier coitarche, unmarried, fewer experienced divorce, a higher proportion of no childbearing, and a higher proportion of current STI. In a cross-sectional observational study of 234 female students in Switzerland, female virgins were less likely to be vaccinated than sexually active women, similar to those who did not express an opinion about the importance of vaccination. The main reasons for refusing vaccination were fear of side effects, parental opposition, and the reluctance of the attending physician [17]. HPV vaccination needs to be initiated before the sex debut. It requires the girls’ health literacy and the understanding and cooperation of their parents, teachers, physicians, and other adults around them for HPV vaccination.

Data on the prevalence of HPV-16/18 infection in the general population of Japanese women are limited. In a phase II clinical trial of a bivalent HPV vaccine in Japan (n = 1040), the prevalence of HPV-16/18 in healthy Japanese women aged 20–25 years was 9.9% at study entry [18]. The results of the Fukui Cervical Cancer Screening Study (n = 7585, age range 25–69 years) showed that the overall prevalence of HPV-16/18 was 1.7% among Japanese women who underwent regular cervical cancer screening [19]. Here, the prevalence of HPV-16/18 among the general Japanese female population aged 16–75 years (median, 30 years) was 5.6%. In 2019, according to the results of a comprehensive survey of living conditions in Japan, the overall cervical cancer screening uptake rate was 43.7%, and women aged 20–24 years had the lowest uptake rate of 15.1% [20]. The cervical cancer screening rate remains quite low in Japan. Therefore, the true prevalence of HPV-16/18 among the general Japanese female population is yet to be determined.

In the present study, the most frequently detected genotype of HPV was HPV-52 (9.1%, 187/2044). We previously conducted an epidemiological survey among Japanese male adults who visited urological clinics and reported that the most frequently detected genotype of HPV was HPV-52 (5.0%, 40/798) [21]. The distribution of the genotypes of HPV among the Japanese female cohort was similar to that of Japanese males. According to the worldwide genotype distribution of HPV in women, HPV-52 was the most prevalent genotype of HPV in Eastern Africa, Japan, and Taiwan [22]; however, HPV-16 was mostly prevalent in other world regions. The ranking of the five most common genotypes of HPV varied across the world regions. In the future, it may be necessary to develop a customized combination of HPV vaccines according to the prevalence and distribution of the HPV genotypes in the world region.

The coronavirus disease 2019 pandemic in Sweden has led to severe difficulties in continuing cervical cancer screenings. Self-sampling testing is a convenient tool for cervical HPV screening, which can overcome some barriers of emotional and practical issues, such as embarrassment, discomfort, and lack of time. The Swedish government switched to vaginal sampling by physicians for self-sampling. Consequently, the population test coverage rate increased by 10%, from 75% to 85%. In July 2022, new government regulations came into effect in Sweden, allowing women to choose whether they want to use a self-sampling kit or be observed by a physician. This has freed up resources so that women coming for their first screening test can also be vaccinated simultaneously. Furthermore, follow-up efforts can be concentrated on women testing positive for cancer-causing HPV infection [23]. If such a drastic change occurs in Japan, we assume that the prevalence of HPV can be revealed easily, and eliminating action against HPV will be more fully realized. The number of countries that employ HPV-based cervical cancer screening by self-sampling has been increasing worldwide and include Argentina, Australia, Denmark, Ecuador, Finland, France, Myanmar, Sweden, Albania, Kenya, Guatemala, Honduras, Malaysia, Netherlands, Peru, Rwanda, and Uganda. The following eight additional countries are piloting self-sampling to decide whether to include this option in their guidelines: Brunei, Mexico, Italy, Spain, El Salvador, Greece, Portugal, and the United Kingdom [24]. Japanese guidelines maintain a conservative stance not to recommend self-sampling because of the limited evidence for HPV testing by self-sampling. Therefore, feasibility studies are required to determine whether self-sampling improves screening uptake or links to the process after a thorough examination [25].

Regarding the efficacy of HPV vaccination, our data suggest that the vaccinated participants were significantly less likely to be infected by HPV-16/18, regardless of the HPV vaccine type or vaccination dose. Kudo and Yamaguchi et al. (2019) reported the effect of bivalent HPV vaccination on 1814 women with a mean age of 20.5 years who received cervical cancer screening in Niigata Prefecture, Japan. Their cohort consisted of three-dose vaccination in 1294 (95.5%), two-dose vaccination in 45 (3.3%), and single-dose vaccination in 16 (1.2%). The odds ratio against HPV-16/18 infection was 0.06 (95% confidence interval, 0.01–0.55) after adjustment for the number of sex partners and fiscal year of birth [26]. We could confirm the effect of HPV vaccination with a larger cohort size by employing detailed characteristics of participants. As described in the results, regarding the association between the age of coitarche and high-risk HPV or HPV-16/18 infections, the odds ratios of the age of coitarche in univariate analyses were <1; however, it was >1 in the multivariate analyses. Participants who initiated sex debut at the age of 20 or older had a longer educational period, fewer smoking habits, fewer lifetime sexual partners, lower experience of divorce, lower commercial sex work experience, and lower rates of a past history of STI. These facts indicate that participants who initiated sex debut at the age of 20 or older are vigilant against STIs, including HPV. However, in the multivariate analyses, participants who initiated sex debut at the age of 20 or older were more likely to be infected by high-risk HPV or HPV-16/18. The significant difference between the participants who initiated sex debut at the age of 20 or older and those who initiated sex debut younger than 20 years was the HPV vaccination rate (23.3% vs. 36.8%; *p* < 0.001). Based on these findings, we believe that HPV vaccine protection is a crucial prerequisite for sexual intercourse. In the Costa Rica HPV Vaccine Trial, single-dose vaccine efficacy against HPV-16/18 infection remained high, and HPV-16/18 antibodies remained stable for more than 11 years after HPV vaccination. A single dose of a bivalent HPV vaccine may induce sufficiently durable protection, which prevents the need for more doses [27]. Moreover, substantial cross-protection was afforded by the bivalent HPV vaccine against HPV-31/33/45. To a lesser extent, HPV-35 and HPV-58 were sustained and remained stable 11 years post-vaccination, reinforcing the notion that a bivalent vaccine is an effective option for protection against HPV-associated cancers [28]. To date, Japan has no national vaccine registry, and official immunization records are managed by each municipal office [26,29]. The Japanese government’s inability to centrally control the HPV vaccination status and the side effects of HPV vaccination may have caused the vaccine crisis in Japan. An Individual Number Card is a plastic card with a portrait and integrated circuit that was first distributed in January 2016 by the Ministry of Internal Affairs and Communications in Japan. Recently, the Digital Agency of Japan released news that the Individual Number Card will be integrated with the Health Insurance Card by the autumn of 2024. If HPV vaccination status is managed with a new Individual Number Card, fewer women may lose the opportunity for vaccination.

This study had some limitations; these included limited information regarding the types of the HPV vaccine, doses of vaccination, and age at vaccination. However, despite such limitations, a single-dose HPV vaccination could reduce the risk of HPV-16/18 infection.

Finally, some evidence for the effectiveness of the HPV vaccine has been currently established. Recent studies in Sweden and England, reported in 2020 and 2021, showed that HPV vaccination in the early teen years decreased the risk of cervical cancer by the age of 30 years by 87% to 88% [30,31]. The Japanese government withdrew a recommendation for HPV vaccination in June 2013 and resumed it in April 2022. The Ministry of Health, Labour and Welfare in Japan also started a catch-up vaccination for women born from April 1997 to April 2006 who lost the opportunity for HPV vaccination in the era of the vaccine crisis.

## 5. Conclusions

Vaccinated women were significantly less likely to be infected by HPV-16/18, regardless of the HPV vaccine type or vaccination dose. Sexually active female citizens who have been unvaccinated need to be protected by HPV vaccines with at least a single dose.

## Figures and Tables

**Figure 1 viruses-15-00159-f001:**
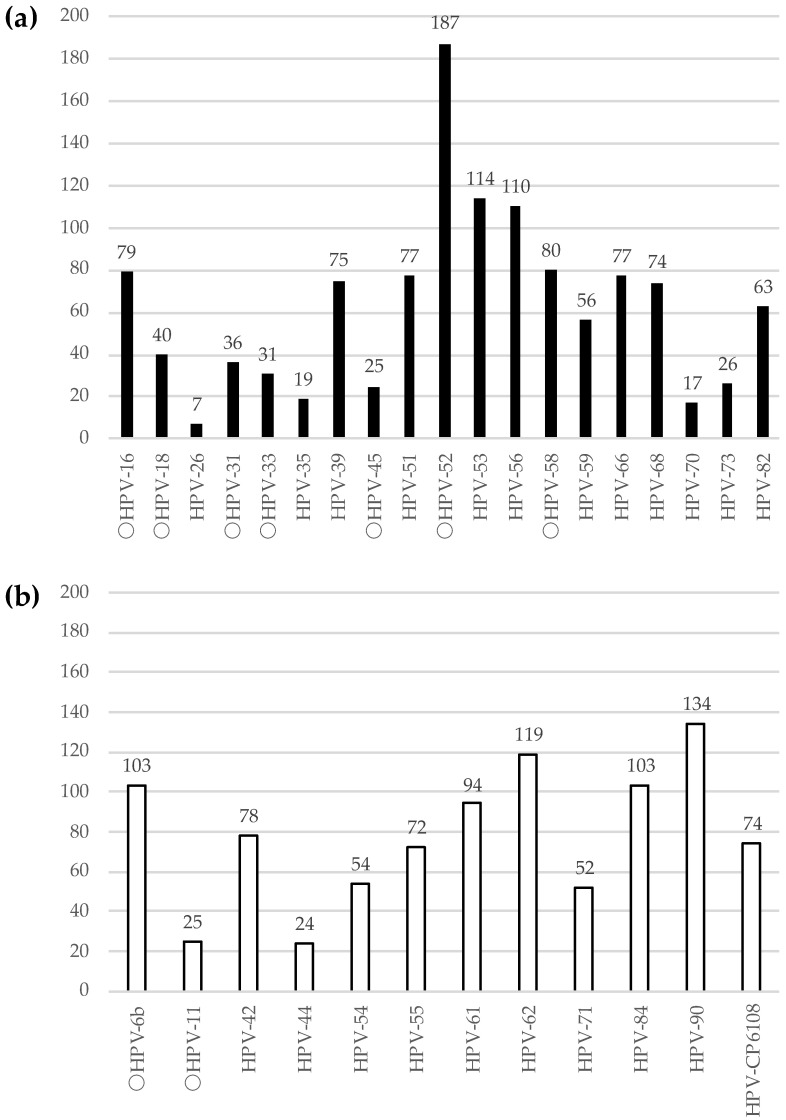
Distribution of HPV genotype. (**a**) High-risk HPV. (**b**) Low-risk HPV. The X-axis indicates the genotypes of HPV. The circle marks at the head of HPV represent the genotypes of HPV covered by the quadrivalent HPV vaccine. The Y-axis indicates the number of participants.

**Figure 2 viruses-15-00159-f002:**
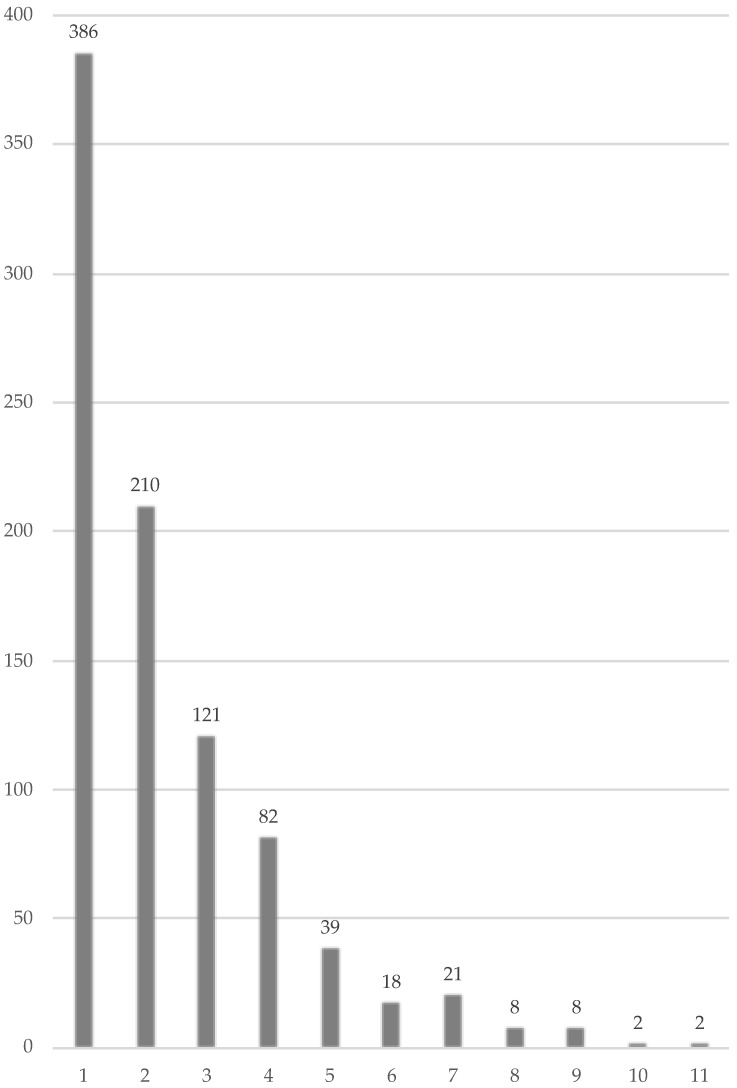
Numbers of participants and HPV genotypes infected. The X-axis indicates the number of genotypes of HPV infected. The Y-axis indicates the number of participants.

**Table 1 viruses-15-00159-t001:** Characteristics of the study participants.

Parameters		Number of Participants
		(Total, *n* = 2044)
Age, year-old	Median (range)	30 (16–75)
Age group, *n* (%)	16–19 years	33 (1.6%)
	20–24 years	423 (20.7%)
	25–29 years	531 (26.0%)
	30–34 years	398 (19.5%)
	35–39 years	270 (13.2%)
	40–44 years	193 (9.4%)
	45–49 years	109 (5.3%)
	≥50 years	87 (4.3%)
Educational status, *n* (%)	≤12 years	294 (14.6%)
	>12 years	1726 (85.4%)
	Missing	24
Smoking status, *n* (%)	Never	1533 (75.7%)
	Former	326 (16.1%)
	Current	165 (8.2%)
	Missing	20
Number of lifetime sex partners, *n* (%)	1–5	994 (50.5%)
	6–10	424 (21.6%)
	11–20	284 (14.4%)
	≥21	265 (13.5%)
	Missing	77
Age of coitarche, *n* (%)	<20 years	1158 (59.9%)
	≥20 years	775 (40.1%)
	Missing	111
Marital status, *n* (%)	Married	742 (36.7%)
	Unmarried	1282 (63.3%)
	Missing	20
Divorce, *n* (%)	Never	1862 (92.3%)
	Once	144 (7.1%)
	Twice	11 (0.5%)
	Missing	27
Number of children, *n* (%)	0	1434 (70.6%)
	1	247 (12.2%)
	2	263 (12.9%)
	≥3	87 (4.3%)
	Missing	13
Experience of commercial sex work, *n* (%)	No	1852 (91.5%)
	Yes	171 (8.5%)
	Missing	21
Current STI, *n* (%)	No	1892 (93.6%)
	Yes	130 (6.4%)
	Missing	22
Past history of STI, *n* (%)	No	1401 (69.5%)
	Yes	615 (30.5%)
	Missing	28
Status of HPV vaccination, *n* (%)	No	1579 (77.7%)
	Yes	454 (22.3%)
	Missing	11

Abbreviations: STI, sexually transmitted infection; HPV, human papillomavirus.

**Table 2 viruses-15-00159-t002:** Status of HPV vaccination and participant characteristics.

Parameters		Unvaccinated	Vaccinated	*p*-Value
		(*n* = 1579)	(*n* = 454)	
Age, years, median (range)		32 (16–75)	24 (18–46)	<0.001
Age group, *n* (%)	16–19	26 (1.6%)	5 (1.1%)	<0.001
	20–24	165 (10.4%)	254 (55.9%)	
	25–29	405 (25.6%)	123 (27.1%)	
	30–34	359 (22.7%)	38 (8.4%)	
	35–39	248 (15.7%)	22 (4.8%)	
	40–44	184 (11.7%)	8 (1.8%)	
	45–49	105 (6.6%)	4 (0.9%)	
	≥50	87 (5.5%)	0 (0.0%)	
Educational status, *n* (%)	≤12 years	252 (16.1%)	42 (9.4%)	<0.001
	>12 years	1316 (83.9%)	407 (90.6%)	
	Missing	11	5	
Smoking status, *n* (%)	Never	1158 (74.0%)	371 (81.7%)	<0.001
	Former	277 (17.7%)	49 (10.8%)	
	Current	130 (8.3%)	34 (7.5%)	
	Missing	14	0	
Number of lifetime sex partners, *n* (%)	1–5	766 (50.2%)	226 (51.7%)	0.703
	6–10	325 (21.3%)	97 (22.2%)	
	11–20	222 (14.5%)	62 (14.2%)	
	≥21	213 (14.0%)	52 (11.9%)	
	Missing	53	17	
Age of coitarche, *n* (%)	<20 years	863 (57.7%)	293 (67.5%)	<0.001
	≥20 years	632 (42.3%)	141 (32.5%)	
	Missing	84	20	
Marital status, *n* (%)	Married	671 (42.8%)	71 (15.7%)	<0.001
	Unmarried	897 (57.2%)	380 (84.3%)	
	Missing	11	3	
Divorce, *n* (%)	Never	1416 (90.7%)	441 (97.8%)	<0.001
	Once	135 (8.6%)	9 (2.0%)	
	Twice	10 (0.6%)	1 (0.2%)	
	Missing	18	3	
Number of children, *n* (%)	0	1017 (64.7%)	412 (90.7%)	<0.001
	1	224 (14.2%)	23 (5.1%)	
	2	251 (16.0%)	12 (2.6%)	
	3	80 (5.1%)	7 (1.5%)	
	Missing	7	0	
Experience of commercial sex work, *n* (%)	No	1434 (91.5%)	414 (91.8%)	0.818
	Yes	134 (8.5%)	37 (8.2%)	
	Missing	11	3	
Current STI, *n* (%)	No	1475 (94.2%)	413 (91.2%)	0.023
	Yes	90 (5.8%)	40 (8.8%)	
	Missing	14	1	
Past history of STI, *n* (%)	No	1085 (69.6%)	313 (69.2%)	0.902
	Yes	475 (30.4%)	139 (30.8%)	
	Missing	19	2	

Abbreviation: STI, sexually transmitted infection; HPV, human papillomavirus.

**Table 3 viruses-15-00159-t003:** Association between the characteristics of the participants and high-risk HPV infection.

Characteristics of the Participants		Crude OR (95% CI)	Adjusted OR * (95%CI)
Age group	16–19	Reference	Reference
	20–24	1.153 (0.553–2.406)	0.991 (0.399–2.457)
	25–29	1.127 (0.543–2.339)	0.897 (0.364–2.208)
	30–34	0.764 (0.364–1.603)	0.671 (0.267–1.686)
	35–39	0.673 (0.316–1.436)	0.586 (0.226–1.517)
	40–44	0.517 (0.236–1.133)	0.445 (0.164–1.210)
	45–49	0.468 (0.201–1.090)	0.395 (0.132–1.183)
	≥50	0.394 (0.161–0.963)	0.872 (0.282–2.701)
Educational status	≤12 years	Reference	Reference
	>12 years	0.765 (0.592–0.989)	1.134 (0.815–1.578)
Smoking status	Never	Reference	Reference
	Former	1.459 (1.137–1.872)	1.294 (0.949–1.763)
	Current	2.292 (1.657–3.170)	1.289 (0.871–1.905)
Number of lifetime sex partners	1–5	Reference	Reference
	6–10	2.941 (2.273–3.806)	2.978 (2.215–4.004)
	11–20	5.239 (3.936–6.974)	5.471 (3.871–7.734)
	≥21	8.718 (6.454–11.775)	8.611 (5.755–12.885)
Age of coitarche	<20 years	Reference	Reference
	≥20 years	0.658 (0.540–0.802)	1.318 (1.015–1.711)
Marital status	Married	Reference	Reference
	Unmarried	2.433 (1.974–3.000)	1.467 (1.046–2.058)
Divorce	Never	Reference	Reference
	Once	1.240 (0.872–1.764)	1.217 (0.773–1.918)
	Twice	5.678 (1.501–21.481)	5.397 (0.926–31.450)
Number of children	0	Reference	Reference
	1	0.565 (0.416–0.768)	0.849 (0.546–1.318)
	2	0.405 (0.293–0.561)	0.677 (0.415–1.103)
	≥3	0.296 (0.163–0.539)	0.590 (0.278–1.252)
Commercial sex work experience	No	Reference	Reference
	Yes	3.089 (2.246–4.249)	0.800 (0.529–1.209)
Current STI	No	Reference	Reference
	Yes	4.193 (2.889–6.086)	2.271 (1.455–3.547)
Past history of STI	No	Reference	Reference
	Yes	3.047 (2.497–3.719)	1.586 (1.225–2.054)
Status of HPV vaccination	No	Reference	Reference
	Yes	1.295 (1.041–1.610)	0.965 (0.717–1.299)

Abbreviations: STI, sexually transmitted infection; HPV, human papillomavirus. * Multivariate analyses were performed for all variables.

**Table 4 viruses-15-00159-t004:** Association between the characteristics of the participants and HPV-16/18 infection.

Characteristics of the Participants		Crude OR (95% CI)	Adjusted OR * (95%CI)
Age group	16–19	Reference	Reference
	20–24	0.419 (0.116–1.509)	0.557 (0.135–2.291)
	25–29	0.793 (0.232–2.714)	0.485 (0.122–1.930)
	30–34	0.670 (0.191–2.349)	0.373 (0.091–1.532)
	35–39	0.506 (0.136–1.877)	0.276 (0.062–1.234)
	40–44	0.604 (0.159–2.294)	0.286 (0.059–1.375)
	45–49	0.283 (0.054–1.475)	0.083 (0.007–0.942)
	≥50	0.482 (0.102–2.280)	0.364 (0.055–2.420)
Educational status	≤12 years	Reference	Reference
	>12 years	0.960 (0.564–1.632)	1.595 (0.843–3.016)
Smoking status	Never	Reference	Reference
	Former	1.332 (0.817–2.171)	1.083 (0.618–1.897)
	Current	1.444 (0.769–2.710)	0.787 (0.362–1.712)
Number of lifetime sex partners	1–5	Reference	Reference
	6–10	3.039 (1.788–5.167)	3.099 (1.694–5.669)
	11–20	2.669 (1.455–4.898)	3.035 (1.514–6.086)
	≥21	5.113 (2.989–8.747)	4.685 (2.282–9.619)
Age of coitarche	<20 years	Reference	Reference
	≥20 years	0.957 (0.645–1.419)	1.560 (0.963–2.527)
Marital status	Married	Reference	Reference
	Unmarried	1.167 (0.782–1.742)	0.879 (0.477–1.619)
Divorce	Never	Reference	Reference
	Once	0.864 (0.394–1.893)	0.722 (0.285–1.827)
	Twice	3.756 (0.801–17.608)	3.452 (0.568–20.986)
Number of children	0	Reference	Reference
	1	1.203 (0.701–2.065)	1.205 (0.562–2.585)
	2	0.643 (0.329–1.257)	0.721 (0.282–1.842)
	≥3	0.581 (0.180–1.878)	0.886 (0.226–3.471)
Commercial sex work experience	No	Reference	Reference
	Yes	2.477 (1.487–4.127)	1.070 (0.557–2.053)
Current STI	No	Reference	Reference
	Yes	3.478 (2.068–5.847)	1.988 (1.045–3.783)
Past history of STI	No	Reference	Reference
	Yes	2.909 (1.986–4.260)	1.798 (1.112–2.908)
Status of HPV vaccination	No	Reference	Reference
	Yes	0.087 (0.028–0.276)	0.047 (0.011–0.196)

Abbreviations: STI, sexually transmitted infection; HPV, human papillomavirus. * Multivariate analyses were performed for all variables.

## Data Availability

Not applicable.

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
