# Peer review of "Prevalence of Human Papillomavirus Types 16/18 and Effect of Vaccination among Japanese Female General Citizens in the Vaccine Crisis Era"

_viruses, 2023, doi:10.3390/v15010159_

Round 1
Reviewer 1 Report
Overall this is an interesting manuscript that explores the prevalence of several high (18) and low-risk HPV strains in a Japanese population. Epidemiological analyses were perform to evaluate the effect of the "HPV vaccine crisis" on HPV occurance.
The data collected is interesting, informative and adds to the body of knowledge. The findings and data will be of interest to the HPV research community.
The following should be addressed prior to publication:
Materials and Methods: It would be helpful to provide additional details on the PCR methodology since the outcomes of this study rely so heavily on the results. What is the read-out on the assay? What are the targets of PCR within the HPV genome. Additional details of the probes used for PCR should be provided in a supplement file.
Results: The authors should consider using graphical representation for some of their data in lieu of exclusively using data tables. The closeness of distribution of various strains would be more apparent.
Were some women infected with multiple strains of HPV? This would be interesting to know.
Discussion and conclusions: The authors do make a case for various epi factors and the incidence of HPV infection.
The authors need to acknowledge and discuss that globally, HPV 16/18 are not necessarily the most common high risk strains of HPV, and discuss this in the context of their findings.
Author Response
Materials and Methods: It would be helpful to provide additional details on the PCR methodology since the outcomes of this study rely so heavily on the results. What is the read-out on the assay? What are the targets of PCR within the HPV genome. Additional details of the probes used for PCR should be provided in a supplement file.
>Thank you for your suggestion. We wrote additional details on the PCR methodology, the target gene of the HPV genome, and the detection system of the PCR products in the Materials and Methods. We cannot provide the details of the probes used for PCR as a supplemental file because we could not get them via the manufacturer’s website.
Results: The authors should consider using graphical representation for some of their data in lieu of exclusively using data tables. The closeness of distribution of various strains would be more apparent.
>We converted Table 3 into Figure 1.
Were some women infected with multiple strains of HPV? This would be interesting to know.
>Of 897 participants who were infected by at least one HPV genotype, 511 female participants were infected by multiple genotypes of HPV. We added sentences of explanation and Figure 2.
Discussion and conclusions: The authors do make a case for various epi factors and the incidence of HPV infection.
The authors need to acknowledge and discuss that globally, HPV 16/18 are not necessarily the most common high risk strains of HPV, and discuss this in the context of their findings.
>Thank you for your critical suggestion. We added some sentences regarding the commonly detected high-risk HPV genotypes in the Discussion.
Reviewer 2 Report
1. Review summary.
The study by Tadaichi Kitamura et al. Aims to elucidate the prevalence and genotype distribution of HPV among Japanese female citizens, and the effect of vaccination against HPV-16/18 in the era of the vaccine crisis. The study correctly addresses the analysis of the data obtained through a survey form and the analysis of viral detection from self-collected vaginal swabs. It is important to distinguish that the authors mention cervical cancer. It seems relevant to me that the authors state how much self-sampling affects viral detection in their study (efficiency and correct sampling). The authors conclude that vaccinated women were significantly less likely to be infected by HPV-16/18, regardless of the HPV vaccine type or vaccination dose. Regarding the above, this reviewer requests that a context be provided in the conclusion and that it be expanded concerning the situation in the country. emphasize genotypes that are not addressed by the bivalent and quadrivalent vaccines.
2. Minor revisions.
1. I recommend the following division in material and methods
- Material collection.
- HPV detection and Genotyping Procedures
- Statistical Methods.
2. DNA extraction should be described into materials and methods.
3. The GENOSEARCH HPV31 kit and the reading equipment must be described in the materials and methods.
4. The authors should clarify whether their methodology detects co-infections
5. Paragraph 194 should be corrected “had not yet had sex were less likely to be vaccinated than sexually active women”
3. Mayor revisions.
1. This reviewer asks the authors to add the significance of the HPV52 positivity found in their study.
2. Authors should include more information on the HPV16 and HPV18 genotypes (Introduction), as they are relevant to their research.

Author Response
Minor revisions.
- I recommend the following division in material and methods
- Material collection.
- HPV detection and Genotyping Procedures
- Statistical Methods.
>Thank you for your suggestion. We added subheadings in the Material and Methods according to your suggestion.
- DNA extraction should be described into materials and methods.
>We added the descriptions concerning DNA extraction in the Materials and Methods.
- The GENOSEARCH HPV31 kit and the reading equipment must be described in the materials and methods.
>We added the descriptions concerning the GENOSEARCH HPV31 kit and the reading equipment in the Materials and Methods.
- The authors should clarify whether their methodology detects co-infections
>Yes. The methodology and equipment we selected can detect co-infections of HPV. We added the explanation in the Material and Methods.
- Paragraph 194 should be corrected “had not yet had sex were less likely to be vaccinated than sexually active women”
>We thought this phrase was correct. We rephrased it as “female virgins were less likely to be vaccinated than sexually active women”.
Mayor revisions.
- This reviewer asks the authors to add the significance of the HPV52 positivity found in their study.
>We added some sentences regarding the HPV-52 positivity in the Discussion.
- Authors should include more information on the HPV16 and HPV18 genotypes (Introduction), as they are relevant to their research.
>We added additional sentences in the Introduction.